

# Nitrous oxide variability at sub-kilometre resolution in the Atlantic sector of the Southern Ocean

Imke Grefe[1,3], Sophie Fielding[2], Karen J. Heywood[1] and Jan Kaiser[1]

[1] Centre for Ocean and Atmospheric Sciences, University of East Anglia, Norwich, United Kingdom
[2] British Antarctic Survey, Cambridge, United Kingdom
[3] Current affiliation: Lancaster Environment Centre, Lancaster University, Lancaster, United Kingdom

## ABSTRACT

The Southern Ocean is an important region for global nitrous oxide ($N_2O$) cycling. The contribution of different source and sink mechanisms is, however, not very well constrained due to a scarcity of seawater data from the area. Here we present high-resolution surface $N_2O$ measurements from the Atlantic sector of the Southern Ocean, taking advantage of a relatively new underway setup allowing for collection of data during transit across mesoscale features such as frontal systems and eddies. Covering a range of different environments and biogeochemical settings, $N_2O$ saturations and sea-to-air fluxes were highly variable: Saturations ranged from 96.5% at the sea ice edge in the Weddell Sea to 126.1% across the Polar Frontal Zone during transit to South Georgia. Negative sea-to-air fluxes ($N_2O$ uptake) of up to $-1.3$ $\mu$mol m$^{-2}$ d$^{-1}$ were observed in the Subantarctic Zone and highest positive fluxes ($N_2O$ emission) of 14.5 $\mu$mol m$^{-2}$ d$^{-1}$ in Stromness Bay, coastal South Georgia. Although $N_2O$ saturations were high in areas of high productivity, no correlation between saturations and chlorophyll $a$ (as a proxy for productivity) was observed. Nevertheless, there is a clear effect of islands and shallow bathymetry on $N_2O$ production as inferred from supersaturations.

## INTRODUCTION

Nitrous oxide ($N_2O$) is a strong greenhouse gas and currently the third largest contributor to radiative forcing after carbon dioxide ($CO_2$) and methane ($CH_4$) (*Hartmann et al., 2013*). Furthermore, $N_2O$ is an important source for stratospheric $NO_x$, which is involved in catalytic ozone depletion (*Crutzen, 1970*; *Ravishankara, Daniel & Portmann, 2009*). The oceans, including coastal zones, estuaries and rivers, are estimated to contribute approximately 25% to global $N_2O$ emissions (*Ciais et al., 2013*). The Southern Ocean alone is estimated to account for 5% of global emissions (0.9 Tg a$^{-1}$ N (nitrogen equivalents), (*Nevison et al., 2005*)). However, measurements of oceanic $N_2O$ concentrations in this region are scarce and these emission estimates are based on atmospheric measurements at Cape Grim and a few seawater measurements in the Southern Ocean from 1977–1993 (*Nevison, Weiss & Erickson, 1995*; *Nevison et al., 2005* and references therein).

Corresponding authors
Imke Grefe, i.grefe@lancaster.ac.uk
Jan Kaiser, J.Kaiser@uea.ac.uk

In the ocean $N_2O$ is produced as a byproduct during nitrification and an intermediate during nitrifier-denitrification in oxygenated waters, and as an intermediate during denitrification in anoxic waters. Nitrification is a two-step process during which ammonia oxidation is followed by nitrite oxidation. The first step is carried out by ammonia oxidising bacteria (AOB) and archaea (AOA). $N_2O$ is a by-product of this step that can be produced during hydroxylamine or nitric oxide oxidation (*Ward, 2008* and references therein). Only recently both steps of nitrification, ammonia oxidation and nitrite oxidation, have been observed in one organism (*Daims et al., 2015*; *Van Kessel et al., 2015*).

Nitrifier-denitrification is the process of ammonia oxidation to nitrite, followed by nitrite reduction via NO and $N_2O$ to $N_2$ (*Wrage et al., 2001*).

Denitrification takes place under anoxic conditions where $NO_3^-$ acts as an electron acceptor in the absence of oxygen ((*Devol, 2008*) and references therein). $NO_3^-$ is reduced to $NO_2^-$, NO and $N_2O$. At the core of oxygen minimum zones, $N_2O$ is further reduced to $N_2$ (*Bange, 2008* and references therein).

The solubility of $N_2O$ in seawater has been described by *Weiss & Price (1980)*. $N_2O$ is a soluble gas, similar to $CO_2$. Due to its high solubility physical processes such as bubble effects do not have a strong effect on surface $N_2O$ saturations. Warming and cooling of surface waters also has only a small effect (4% per degree centigrade). Due to the non-linearity of saturation curves for water mass mixing, diapycnal mixing, upwelling and entrainment of deeper water masses into the surface only has an impact on $N_2O$ saturations if there is a large concentration gradient. Examples for high $N_2O$ surface saturations due to these processes are rising isopycnals at the Southern Antarctic Circumpolar Current Front (SACCF, see below, *Chen et al., 2014* and this work) and upwelling along the California coast (*Lueker et al., 2003*).

The Scotia Sea in the Atlantic sector of the Southern Ocean is confined by the Scotia Ridge to the north, east and south and the Drake Passage to the west (*Atkinson et al., 2001*). South Georgia is part of the North Scotia Ridge in the path of the Antarctic Circumpolar Current (ACC) and within the Antarctic Zone (AAZ) south of the Polar Front (PF).

The ACC is driven by strong, westerly winds between approximately 45–55°S (*Trenberth, Large & Olson, 1990*) and connects the Atlantic, Pacific and Indian Ocean. Isopycnals rise step-wise towards the south, characterising the ACC fronts and their resulting strong surface currents (*Orsi, Whitworth & Nowlin, 1995* and references within): the Subantarctic Front (SAF), Polar Front (PF) and the Southern ACC Front (SACCF), accounting for most of the ACC transport. The SAF is characterised by the rapid subduction of the salinity minimum of Antarctic Intermediate Water (AAIW) to the north (*Whitworth & Nowlin, 1987*). Subduction of the temperature minimum of the AASW marks the position of the PF (*Gordon, 1967*). The SACCF is then characterised by a subsurface horizontal temperature gradient as Upper Circumpolar Deep Water (UCDW) shoals to the south. The Southern Boundary of the ACC (SB) can be identified by a loss of UCDW properties (temperature, salinity, oxygen minimum) to the south. To the north of the SB the UCDW isopycnal shoals and UCDW can be entrained into the Surface Mixed Layer (SML) (*Orsi, Whitworth & Nowlin, 1995*). The Weddell-Scotia Confluence separates the ACC from the waters of the Weddell Gyre (*Gordon, 1967*) and is a regional feature of the Scotia Sea.

Around South Georgia, the path of the ACC is strongly influenced by the bathymetry (*Atkinson et al., 2001* and references therein): The South Georgia Ridge to the west of the island deflects the ACC to the north, resulting in strong large- and mesoscale variability. Additionally, the wide shelf seems to influence temperature and salinity in the waters around the island. It was observed that in summer the water is slightly warmer and fresher than in the surrounding ocean, possibly due to solar heating of the shallow water column and local runoff, respectively (*Priddle, Heywood & Theriot, 1986*; *Brandon et al., 2000*).

Waters around South Georgia are characterised by high abundances of phytoplankton, zooplankton and vertebrate predators (*Atkinson et al., 2001*), whereas most other areas of the Southern Ocean are dominated by High Nutrient Low Chlorophyll (HNLC) conditions (*Martin, 1990*). A relatively stable water column and benthic iron input support productivity in the vicinity of the island (*Holeton et al., 2005*; *Korb et al., 2005*).

To the south of the Scotia Sea and the ACC, the Weddell Sea is characterised by a large gyre, flowing eastwards until 20–30°E, returning westwards along the continental margin and following the eastern coast of the Antarctic Peninsula northwards (*Deacon, 1979*). Seasonal blooms in the Weddell Sea are associated with the Antarctic shelf and the ice edge (*El-Sayed & Taguchi, 1981*; *Kristiansen, Syvertsen & Farbrot, 1992*; *Nelson et al., 1989*; *Smith Jr & Nelson, 1990*). Furthermore, drifting icebergs stimulate productivity by input of terrigenous iron through melt water (*Biddle et al., 2015*; *Smith et al., 2007*).

Generally, the Southern Ocean has the potential for both production and removal of $N_2O$ (*Rees, Owens & Upstill-Goddard, 1997*): Solubility of $N_2O$ increases at lower temperatures, and together with downwelling areas associated with deep-water formation and convergences in the Antarctic frontal zones, wide areas could function as sinks. On the other hand, upwelling of deep and intermediate waters could be a source of biologically produced $N_2O$ to the atmosphere. Artificial iron fertilisation of the Southern Ocean may stimulate biological $CO_2$ uptake, but could potentially increase $N_2O$ production, offsetting the benefits of $CO_2$ sequestration in terms of radiative forcing (*Fuhrman & Capone, 1991*; *Jin & Gruber, 2003*). While an iron fertilisation experiment in the Australasian sector of the Southern Ocean showed $N_2O$ accumulation in the pycnocline (*Law & Ling, 2001*), no increase in $N_2O$ concentrations was observed during a similar experiment in the Atlantic subpolar sector (*Walter et al., 2005*).

Here, we present high-resolution measurements of ocean surface $N_2O$ concentrations from the Scotia Sea and Weddell Sea. Using laser off-axis Integrated Cavity Output Spectroscopy (ICOS) (*Arevalo-Martinez et al., 2013*; *Grefe & Kaiser, 2014*) combined with wind-speed gas exchange parameterisations we can resolve small-scale variability in $N_2O$ fluxes and capture the impact of frontal structures and changes in weather conditions on emissions from the Southern Ocean.

## METHODS

$N_2O$ concentrations in surface waters were measured during the annual Western Core Box (WCB) krill survey in the Scotia Sea between 28 December 2011 and 16 January 2012 (JR260B) and in the Weddell Sea from 20 January to 2 February 2012 (JR255A/GENTOO

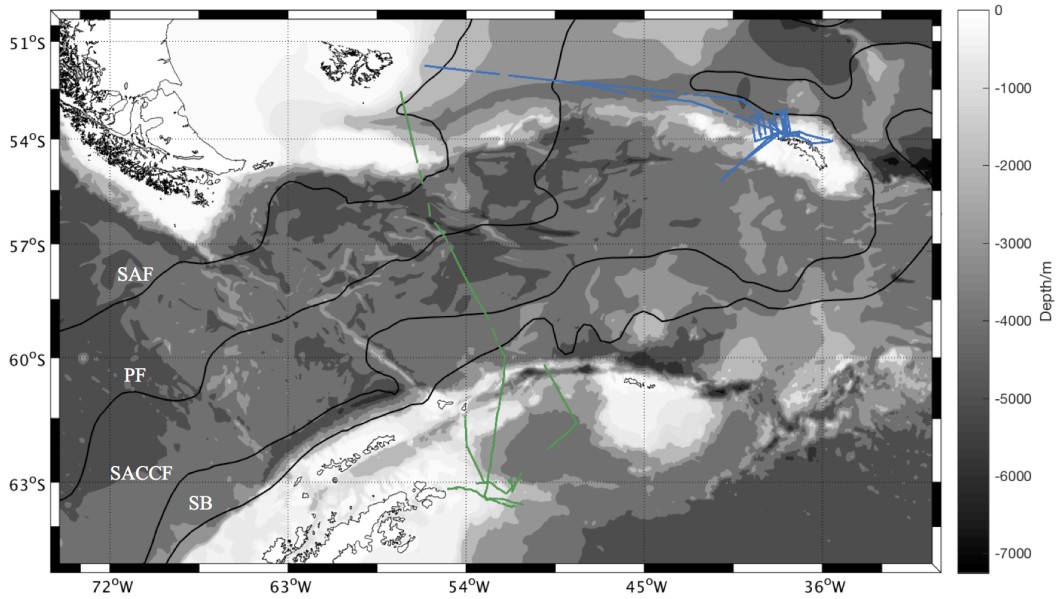

**Figure 1  Overview of sampling area.** Sampling area with bathymetry from the General Bathymetric Chart of the Oceans (GEBCO) one minute grid (*Jakobsson et al., 2008*). Isobaths every 1,000 m between 8,000 m and 1,000 m depth, every 200 m between 1,000 m and 0 m. $N_2O$ measurement positions during JR260B in blue, $N_2O$ measurement positions during JR255A/GENTOO in green. Climatological locations of fronts in black given in *Orsi, Whitworth & Nowlin (1995)*: SAF Sub-Antarctic Front, PF Polar Front, SACCF Southern Antarctic Circumpolar Current Front, SB Southern Boundary of the Antarctic Circumpolar Current.

Gliders: Excellent New Tools for Observing the Ocean) on board RRS *James Clark Ross*. The measurement region for both cruises is shown in Fig. 1.

The setup and performance of the coupled analyser-equilibrator system is described in *Grefe & Kaiser (2014)*. Briefly, a percolating glass bed equilibrator was connected to a $N_2O/CO$ analyser ($N_2O/CO$-23d, Los Gatos Research Inc.). Artificial air mixtures (21% $O_2$, 79% $N_2$) with nominal $N_2O$ mole fractions of 300, 320 and 340 nmol mol$^{-1}$ (BOC) were used as reference gases. These gas mixtures were compared with IMECC/NOAA standards to determine the exact values of $(297.6 \pm 0.1)$, $(325.3 \pm 0.1)$ and $(344.2 \pm 0.1)$ nmol mol$^{-1}$ (NOAA-2006 scale). Reference gases and marine background air were measured twice a day for 20 min each during JR260B. Due to sufficient analyser stability during this cruise, calibration was reduced to once a day during JR255A/GENTOO. To ensure complete flushing of the cavity, only the last 5 min of each 20-min gas measurement were evaluated. Correspondingly, the first 15 min of equilibrator data following the reference gas and air measurements were discarded. Precision for single reference measurements (standard deviation of measurements at 1 Hz over 5 min) was 0.4 nmol mol$^{-1}$. The standard deviation of the uncorrected reference gas measurements throughout JR260B was 1.1 nmol mol$^{-1}$ ($n = 29$) and 0.8 nmol mol$^{-1}$ ($n = 19$) for JR255A/GENTOO. To estimate the long-term repeatability of the measurements after calibration, we used the standards with the lowest and highest $N_2O$ mole fraction to calibrate the standard with the middle

$N_2O$ mole fraction. The mean difference between resulting calibrated $N_2O$ mole fraction and the actual value of 325.3 nmol $mol^{-1}$ was $(0.2 \pm 0.1)$ nmol $mol^{-1}$, corresponding to a precision better than 0.1%. The flow rate of the headspace gas through the analyser was 400 mL $min^{-1}$ (293 K, 1 bar) resulting in a 95% relaxation time ($t_{95} = 3\tau$) of approximately 7 min (*Grefe & Kaiser, 2014*). Data were acquired at a rate of 1 Hz; which were binned into 60 s averages for data evaluation purposes. Water flow through the equilibrator was held constant at 1.8–1.9 L $min^{-1}$, using a flow regulator ($\frac{1}{2}$ inch diameter tap tail flow regulator; Robert Pearson & Company Ltd, Warminster, UK). Water temperature in the equilibrator was measured with a Pt-100 temperature probe (Omega Engineering Limited, Manchester, UK), calibrated against a mercury reference thermometer to within $\pm 0.06\,°C$, and recorded using a RTD temperature recorder and USB Datalogger interface (both Omega Engineering Limited).

Concentration of dissolved $N_2O$ ($c$) was calculated from the dry mole fraction measured in the equilibrator headspace ($x$), water temperature in the equilibrator ($T_{eq}$), salinity ($S$) and atmospheric pressure ($p_{air}$) using the solubility function $F$ as described by *Weiss & Price (1980)*, Eq. (1).

$$c = xF\left(T_{eq}, S\right)p_{air}. \tag{1}$$

$N_2O$ saturation in surface waters ($s$) was calculated by comparing $x$ with the atmospheric mole fraction $x_{air}$ and the respective equilibrium concentrations for ($T_{eq}$) and temperature at the seawater intake ($T_{in}$) Eq. (2).

$$s = \frac{xF\left(T_{eq}, S\right)}{x_{air}F\left(T_{in}, S\right)}. \tag{2}$$

Values for $p_{air}$, S and $T_{in}$ were from the ship's surface water and meteorological monitoring system Surfmet (http://www.bodc.ac.uk).

Sea-to-air flux was calculated using wind speeds at 10 m above sea level from the CCMP Wind Vector Analysis Product (http://www.remss.com/measurements/ccmp). CCMP data have been interpolated to match cruise dates and positions for JR260B and JR255A/GENTOO. The gas transfer coefficient ($k_w$) was calculated, using the parameterisation of *Nightingale et al. (2000)* (Eq. (3)) and $k_w$ was adjusted for $N_2O$ with the Schmidt number $Sc$ calculated following (*Wanninkhof, 2014*), Eq. (3).

$$\frac{k_w}{m\ d^{-1}} = 0.24\left[0.222\left(\frac{u_{CCMP}}{m\ s^{-1}}\right)^2 + 0.333\frac{u_{CCMP}}{m\ s^{-1}}\right]\left(\frac{Sc}{600}\right)^{-0.5}. \tag{3}$$

The air-sea flux ($\Phi$) was calculated from $k_w$ and the difference between $N_2O$ concentrations in seawater $c$ and air saturation concentrations ($c_{sat}$), Eq. (4):

$$\Phi = k_w\left(c - c_{air}\right) = k_w\left[c - x_{air}F\left(T_{in}, S\right)p_{air}\right]. \tag{4}$$

Ship-based underway measurements of sea surface temperature and salinity were used to identify frontal structures for JR260B and JR255A/GENTOO. The location of these areas was then compared to the climatological front positions of *Orsi, Whitworth & Nowlin (1995)* to ensure positions based on measured salinity and temperature fall within expected geographical locations.

Satellite products for chlorophyll $a$ and climatology data for nitrate ($NO_3^-$) concentrations were compared with *in situ* $N_2O$ saturation data. Chlorophyll $a$ concentrations were obtained from MODIS-Aqua (https://oceandata.sci.gsfc.nasa.gov) for December 2011 and January 2012. The level-3 product was downloaded at a 4 km resolution. Climatological mean dissolved $NO_3^-$ concentrations were downloaded from the World Ocean Atlas 2013 (*Garcia et al., 2013*) for boreal winter (https://www.nodc.noaa.gov/OC5/woa13/woa13data.html). This climatology data was gridded 1° longitude by 1° latitude for 37 standard depths. Depth profiles taken during station work south of the PF and ACC for JR260B and JR255A/GENTOO respectively suggested a mixed layer depth (MLD) of approximately 60 m, similar to MLD derived from Argo float profiles January to March (*Dong et al., 2008*). $NO_3^-$ concentrations from this depth were extracted from the climatology and interpolated along the cruise track for comparison with $N_2O$ saturations.

# RESULTS AND DISCUSSION

## $N_2O$ concentrations and saturations in the surface ocean

$N_2O$ dry mole fractions measured in marine air were on average $(323.6 \pm 0.6)$ nmol mol$^{-1}$ during JR260B and $(324.0 \pm 0.7)$ nmol mol$^{-1}$ during JR255A/GENTOO. These values agree within measurement uncertainties with data from the Advanced Global Atmospheric Gases Experiment (AGAGE) for Cape Grim, Tasmania in January 2012 (40.68°S 144.69°E, $(323.9 \pm 0.5)$ nmol mol$^{-1}$). Surface water saturations and sea-to-air-flux were calculated based on the mean atmospheric mole fractions measured during JR260B and JR255A/GENTOO.

### JR260B

Concentrations and saturations of $N_2O$ in the surface ocean for JR260B are shown in Figs. 2A and 2B, respectively. Average $N_2O$ concentrations were $(14.0 \pm 0.7)$ nmol L$^{-1}$, ranging from 11.2 nmol L$^{-1}$ to 16.2 nmol L$^{-1}$. Average $N_2O$ saturation was $(104.3 \pm 3.4)$%, ranging from 99.4% to 126.1%.

For the purposes of the discussion of the data, they were divided into four different areas based on differences in biogeochemical conditions and oceanographic settings:

(a) Measurements between the Falkland Islands and the SAF in comparatively warm, saline surface waters ($\theta$ >7 °C, S 34.1–34.2) were located in the *Subantarctic Zone (SAZ)*.

(b) The *Polar Frontal Zone (PFZ)* covered the rapidly cooling and freshening surface waters ($\theta$ declining from 7 to 3 °C, S declining from 34.2 to 33.9) between the SAF and PF.

(c) The *Antarctic Zone (AAZ)* was identified by the relatively homogeneous cold and fresh *T-S* signature of AASW.

(d) *Stromness Bay* is discussed separately, due to the unique influence of the island of South Georgia in this area.

*Subantarctic Zone (SAZ).* In the Subantarctic Surface Waters (SASW) close to the Falkland Islands to the east of the Subantarctic Front (SAF), average surface concentrations were low, while saturations were similar to those observed in the Antarctic Zone (AAZ) $(11.4 \pm 0.2)$ nmol L$^{-1}$, $(102.4 \pm 0.7)$%, see below). Water temperature and salinity were higher

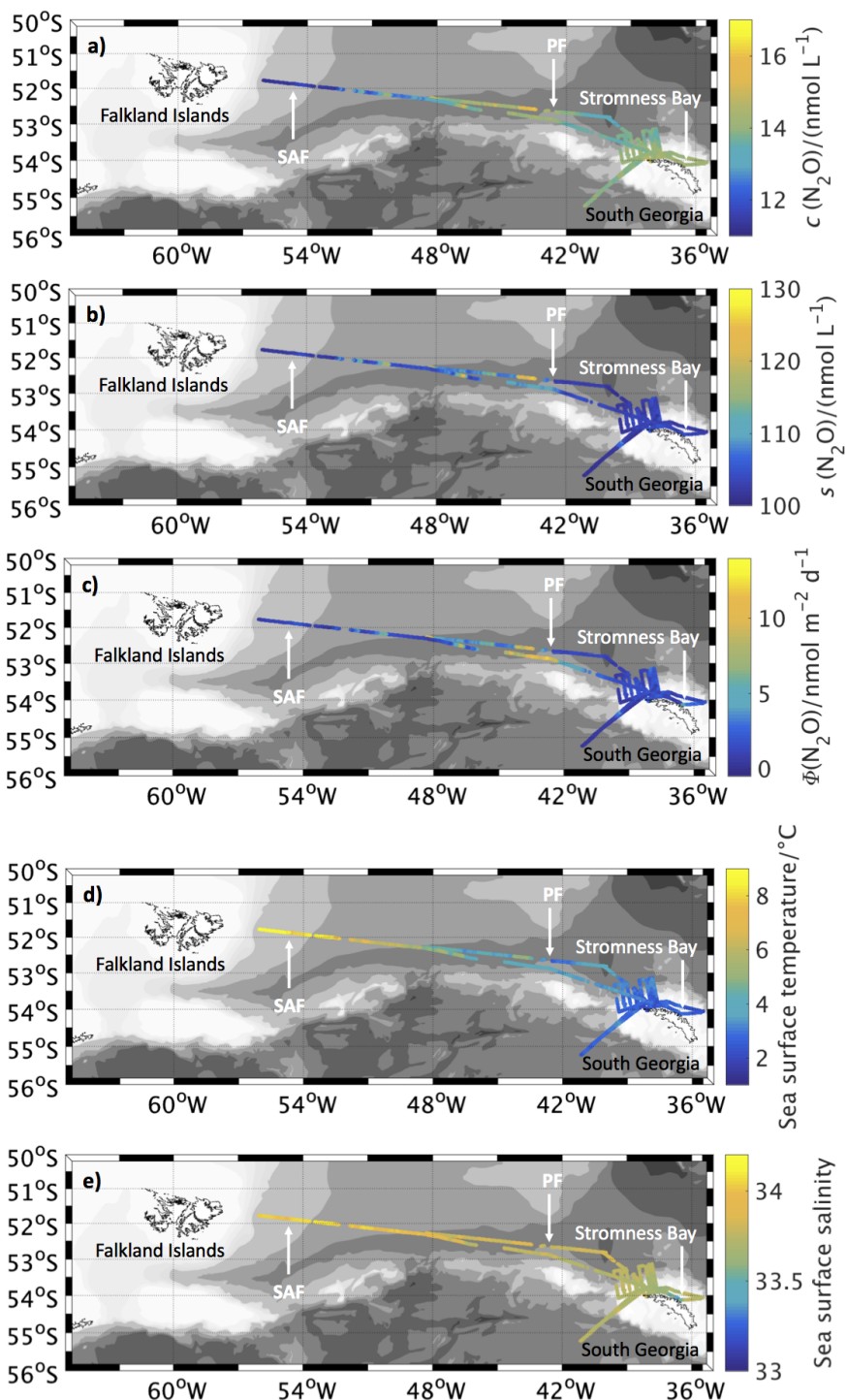

**Figure 2  N$_2$O surface concentrations, saturations and fluxes, as well as sea surface temperature and salinity along cruise track for JR260B.**  (A) N$_2$O concentration in surface waters along the cruise track of JR260B. (B) N$_2$O saturations calculated from measured atmospheric and seawater dry mole fractions. (C) Sea-to-air flux along the cruise track of JR260B. (D) Sea surface temperature and (E) salinity from underway measurements. Approximate position of Subantarctic Front (SAF) and Polar Front (PF) are indicated, based on sea surface temperature and salinity measurements during JR260B.

than in the Antarctic waters south and east of the PF, so the difference in concentrations was mostly due to solubility effects. *Zhan & Chen (2009)* observed higher saturations (110 to 135%) to the north of the SAF in the Indian Sector of the Southern Ocean. The cause of this difference between the two sectors of the Southern Ocean is not known and highlights the need for more research to better understand marine $N_2O$ cycling.

*Polar Frontal Zone (PFZ).* $N_2O$ concentrations across the PFZ between the SAF and PF were highly variable with values ranging from 11.4 to 16.2 nmol $L^{-1}$ (101.7 to 126.1% saturation) while water temperature and salinity were decreasing towards the east. Average concentrations and saturations across the PFZ were (13.8 $\pm$ 0.9) nmol $L^{-1}$ and (108.8 $\pm$ 5.2)%, respectively.

The most important sources of $N_2O$ to surface waters are upwelling of old water masses with preformed high $N_2O$ concentrations and *in situ* production during remineralisation/nitrification or denitrification. High $N_2O$ saturations across the PFZ could neither be attributed to a specific water mass (identifiable from temperature, salinity and density of the surface water), nor to a point on the mixing line between Subantarctic Surface Water (SASW) and Antarctic Surface Water (AASW) across the PFZ (Fig. 3A). Therefore, it is unlikely that upwelling and mixing processes or regional oceanographic features are the source of the high $N_2O$ saturations across the PFZ.

Frontal systems can supply nutrients and iron to surface waters with large phytoplankton blooms forming across the PFZ, potentially fuelled by iron input from the Antarctic Peninsula archipelago, the Scotia Ridge and Georgia Rise (*De Baar et al., 1995*; *Wadley, Jickells & Heywood, 2014*). Satellite observations showed high chlorophyll *a* concentrations in proximity to high $N_2O$ supersaturations in the PFZ (Fig. 4A, 'Comparison of $N_2O$ surface saturations to chlorophyll a and nitrate concentration'). Remineralisation of sinking bloom biomass could lead to enhanced *in situ* $N_2O$ production across the PFZ, resulting in high saturations.

In contrast to the high surface $N_2O$ concentrations and saturations in the PFZ during JR260B, data from the Indian sector of the Southern Ocean showed minimum $N_2O$ inventories within the PFZ (*Farías et al., 2015*). Farías et al. identify uptake of $N_2O$ as an alternative substrate during biological nitrogen fixation as a potential sink process, presumably stimulated by iron input from the sediments of the Kerguelen archipelago. While both islands, South Georgia in the Atlantic and the Kerguelen Islands in the Indian Sector, are positioned within the PFZ and sediments are a source of iron to the water column, the data from the Kerguelen region were collected to the west of the islands, whereas JR260B data were collected to the east of South Georgia. The geographic location of the measurements in relation to islands could affect the biogeochemical properties of the water column and the community composition of $N_2O$ cycling bacteria and archaea (*Santoro et al., 2011*; *Löscher et al., 2012*) and therefore result in major differences in the local source–sink behaviour of the ocean.

*Antarctic Zone (AAZ).* Average $N_2O$ concentrations in the AAZ to the west of the PF near South Georgia were substantially higher than in the SAZ (14.1 $\pm$ 0.4) nmol $L^{-1}$. Saturations,

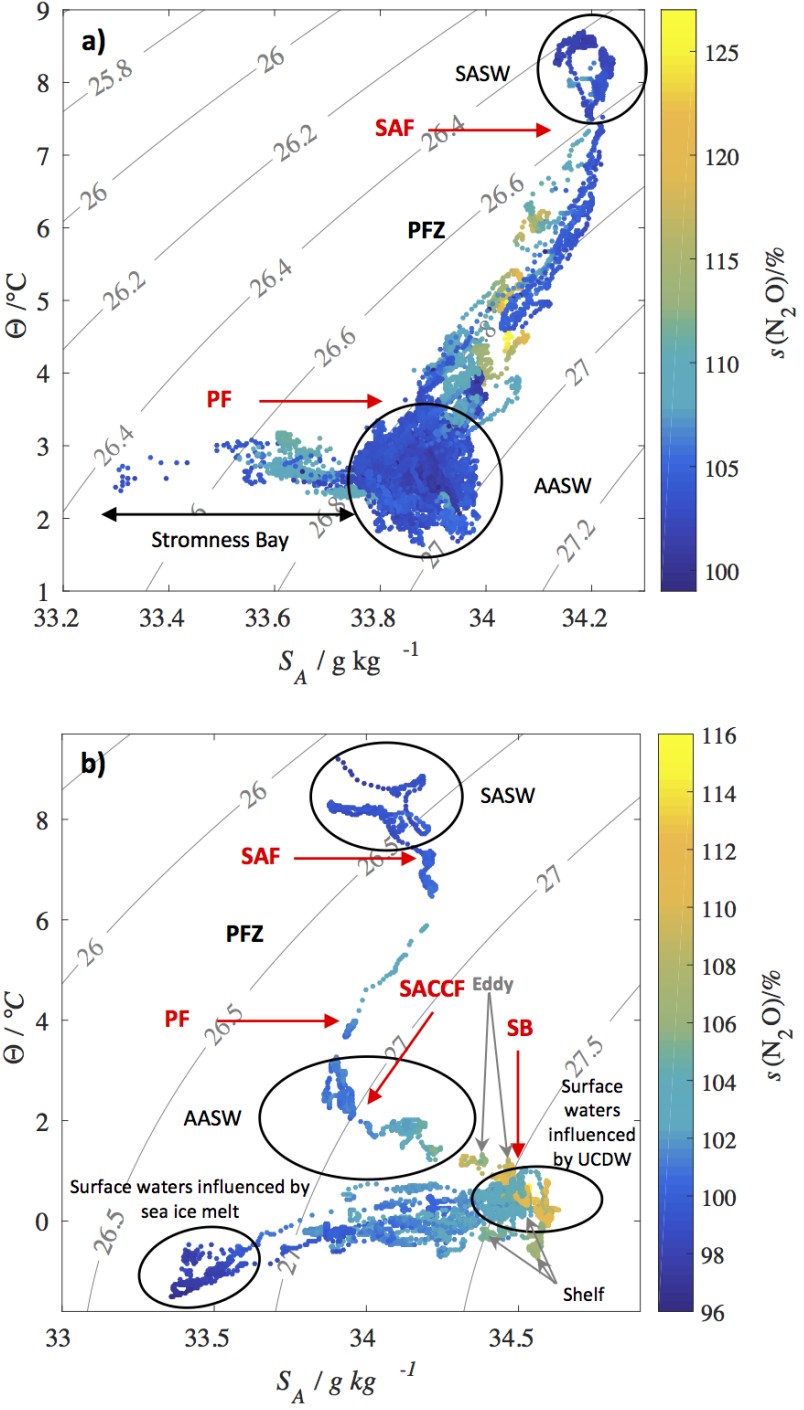

**Figure 3 N₂O saturations along potential density contours.** N₂O saturations $s(N_2O)$ with corresponding absolute salinities $S_A$ and conservative temperatures $\Theta$ for (A) JR260B and (B) JR255A/GENTOO. Contour lines are the potential density anomalies ($\sigma_\theta$) calculated using the TEOS-10 GSW toolbox (*McDougall & Barker, 2011*). Approximate positions of the Subantarctic Front (SAF), Polar Front (PF), Southern Antarctic Circumpolar Front (SACCF) and Southern Boundary of the Antarctic Circumpolar Current (SB) based on sea surface temperature and salinity measurements are indicated by red arrows. SASW, Subantarctic Surface Water; AASW, Antarctic Surface Water; UCDW, Upper Circumpolar Deep Water.

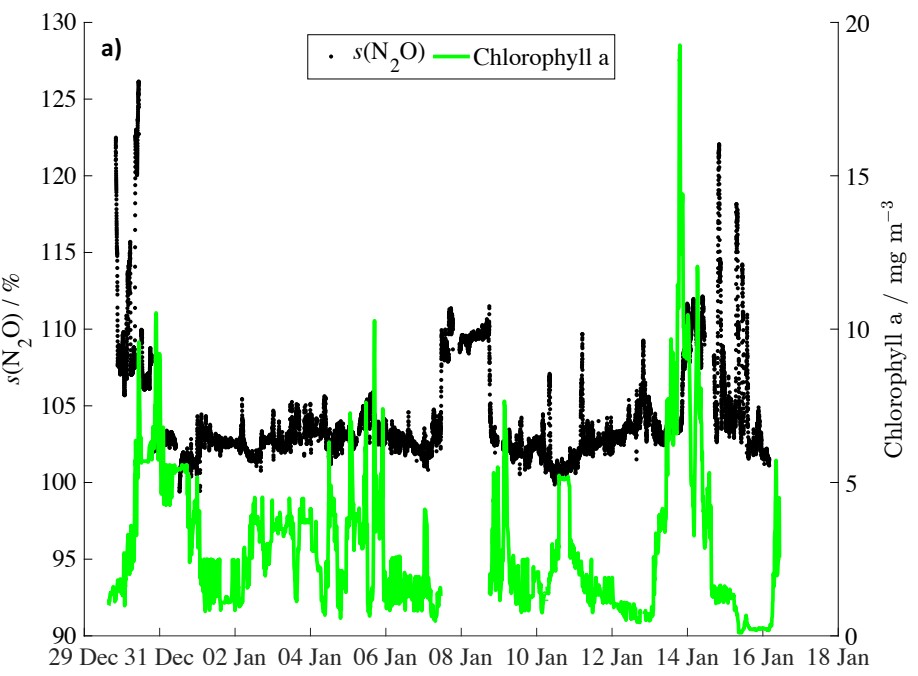

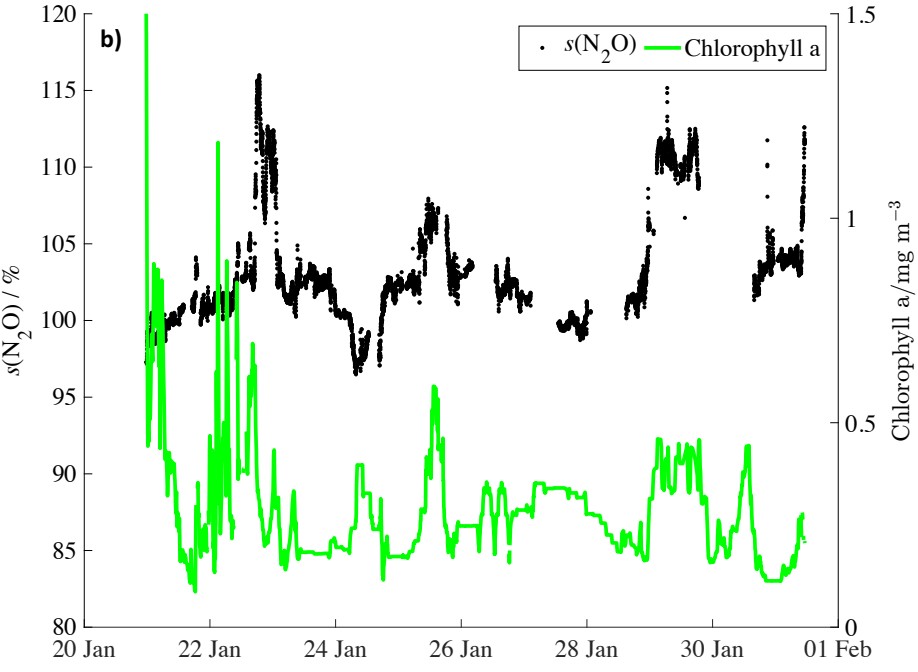

**Figure 4** **N$_2$O saturations and chlorophyll concentrations for JR260B and JR255A/GENTOO.** Surface N$_2$O saturation during (A) JR260B (December 2011) and (B) JR255A/GENTOO (January 2012) in black and average chlorophyll concentrations from MODIS-Aqua in green. Due to cloud cover, there are no satellite data in the vicinity of South Georgia (8 Jan). The co-located $s$(N$_2$O) and $c$(Chl $a$) increases on 25 and 29 January correspond to the Antarctic shelf and the standing eddy (section 'JR255A/GENTOO'), respectively.

however, were only slightly increased to $(103.9 \pm 3.1)\%$, similar to values of $(103.0 \pm 2.0)\%$ previously observed in the region (*Weiss et al., 1992*). The lower water temperature and salinity of Antarctic Surface Water AASW increased $N_2O$ solubility, resulting in lower saturation values. Upwelling of deep water masses would not be expected away from the ACC and denitrification is not likely to take place in the oxygenated surface waters of the AAZ. Nitrification and nitrifier-denitrification would be more likely sources of *in situ* $N_2O$ production. High nitrification rates ($>30$ mmol $m^{-2}$ $d^{-1}$) were suggested for the Pacific sector of the Southern Ocean south of the PF (*Sambrotto & Mace, 2000*) and could also account for $N_2O$ supersaturation observed in the AAZ of the Atlantic sector. Extensive phytoplankton blooms were observed in the vicinity of South Georgia and across the Scotia Sea, extending to the southern limit of the Polar Front (*Korb, Whitehouse & Ward, 2004*; *Korb et al., 2005*). These blooms can develop due to a stable water column over the shelf and iron input from the shelf sediments and island runoff (*De Baar et al., 1995*; *Holeton et al., 2005*; *Korb et al., 2005*; *Wadley, Jickells & Heywood, 2014*). This accumulation of biomass would supply ample substrate to sustain high nitrification rates and $N_2O$ production could lead to the observed supersaturation close to the island, despite the absence of a clear correlation between chlorophyll a, climatological $NO_3^-$ concentrations and $N_2O$ saturations ('$N_2O$ sea-to-air flux'). Not much is known about nitrifier-denitrification in polar waters and additional research is required to estimate the importance of this production pathway for $N_2O$ accumulations in surface waters.

*Stromness bay.* $N_2O$ concentrations and saturations in coastal Stromness Bay, South Georgia, were higher than in the open waters of the AAZ (($14.8 \pm 0.3$) nmol $L^{-1}$ and ($108.1 \pm 2.6$)%, respectively) with highest saturations observed where fresh meltwater runoff from the island was mixing with AASW (Fig. 3A). Terrestrial runoff can transport iron and biomass from land into the sea and stimulate productivity and subsequent remineralisation where $N_2O$ is produced during nitrification. In addition, fur seals (*Arctocephalus gazella*) and macaroni penguins (*Eudyptes chrysolophus*) have large breeding colonies on South Georgia, re-distributing nitrogen from their hunting grounds to the island (*Whitehouse et al., 1999*).

While no $NO_3^-$ concentrations are available for JR260B, *Rees et al. (2016)* observed high $NO_3^-$ values of 25.8 μmol $L^{-1}$ at the base of the mixed layer north of South Georgia in austral summer 2013. Concentrations would be expected to be even higher in Stromness Bay due to land run-off and the high abundance of penguins and seals observed at the shore and in the water at the time of measurements. High nitrogen loads in coastal waters are expected to lead to high $N_2O$ saturations (*Bange, Rapsomanikis & Andreae, 1996*) and would explain the consistently high supersaturations observed in Stromness Bay during JR260B.

Nitrifier-denitrification in anoxic sites of suspended particles can be an additional source of $N_2O$ (*Ostrom et al., 2000*). As the water depth in Stromness Bay is shallow (60 m) and well mixed at the anchoring site, $N_2O$ produced by denitrification at the sediment interface could have diffused into the water column, contributing to the high concentration at the surface.

### JR255A/GENTOO

During JR255A/GENTOO, average $N_2O$ surface concentrations were $(14.9 \pm 1.2)$ nmol $L^{-1}$, corresponding to saturations of $(103.1 \pm 3.6)\%$ (range: $(10.6–17.0)$nmol $L^{-1}$; $(96.5–116.0)\%$ saturation). While $N_2O$ concentrations were higher than for JR260B, mainly due to the lower water temperatures, average saturations were slightly lower.

As for JR260B, data have been discussed for three areas, based on substantial differences in biogeochemical conditions and oceanographic settings:

(a)  Data collected between the Falkland Islands and the South Scotia Ridge were discussed below as *Drake Passage*, crossing the SAF ($\theta$ >7 °C to the north), PF ($\theta$ decreasing from 7 °C to 3 °C, S from 34.2 to 33.8), SAACF (identified by a small salinity change in surface waters from 33.8 to 34) and SB (surface waters influenced by UCDW, S increases from 34.4 to 34.6);

(b)  South of the South Scotia Ridge, measurements were designated as *Weddell Sea* data, including a standing eddy (identified by a decrease in surface salinity from 34.4 to 34.2 and an increase of surface temperature at the edges from 0.5 to 1 °C, *Thompson et al., 2009*) forming over the ridge and data collected from the Antarctic shelf;

(c)  Measurements influenced by the *sea ice edge* in the Weddell Sea ($\theta$ <0 °C and salinity <33.6).

*Drake Passage.*  Average $N_2O$ concentrations at the beginning of the cruise were $(13.7 \pm 1.8)$ nmol $L^{-1}$ $((102.9 \pm 4.2)\%$ saturation). Decreasing temperatures during transit across Drake Passage to the Eastern Antarctic Peninsula partially accounted for increasing $N_2O$ concentrations (Fig. 5A), resulting in a wide range of concentrations across the region, whereas saturations were on average only slightly above equilibrium with the atmosphere. Water masses across the Drake Passage were Subantarctic Surface Water to the north of the SAF and Antarctic Surface Water to the south of the PF (SASW and AASW, respectively) (Fig. 3B). As these surface waters are in constant contact with the atmosphere, they are expected to be in equilibrium with the atmosphere in absence of biological $N_2O$ sources. $N_2O$ saturations above 100% would therefore be due to *in situ* production during remineralisation of biomass.

For comparison, *Rees, Owens & Upstill-Goddard (1997)* observed slightly lower saturations for Drake Passage $((99.7 \pm 3.0)\%)$, while *Weiss et al. (1992)* reported values close to the data presented here (Ajax 2: $(102.3 \pm 0.9)\%$, JR255A/GENTOO: $(102.9 \pm 4.2)\%$). Higher saturation values for *Weiss et al. (1992)* and JR255A/GENTOO could be due to interannual and seasonal variability. JR255A/GENTOO and Ajax 2 took place in January and February, later in the austral summer season than the November/December cruise of *Rees, Owens & Upstill-Goddard (1997)*. Nitrification as part of the remineralisation of sinking biomass could have increased $N_2O$ accumulations in the mixed layer over the summer, resulting in higher values later in the growing season. No direct link to chlorophyll *a* concentrations was found ('Comparison of $N_2O$ surface saturations to chlorophyll a and nitrate concentrations'), and was unlikely to exist due to the temporal and spatial decoupling.

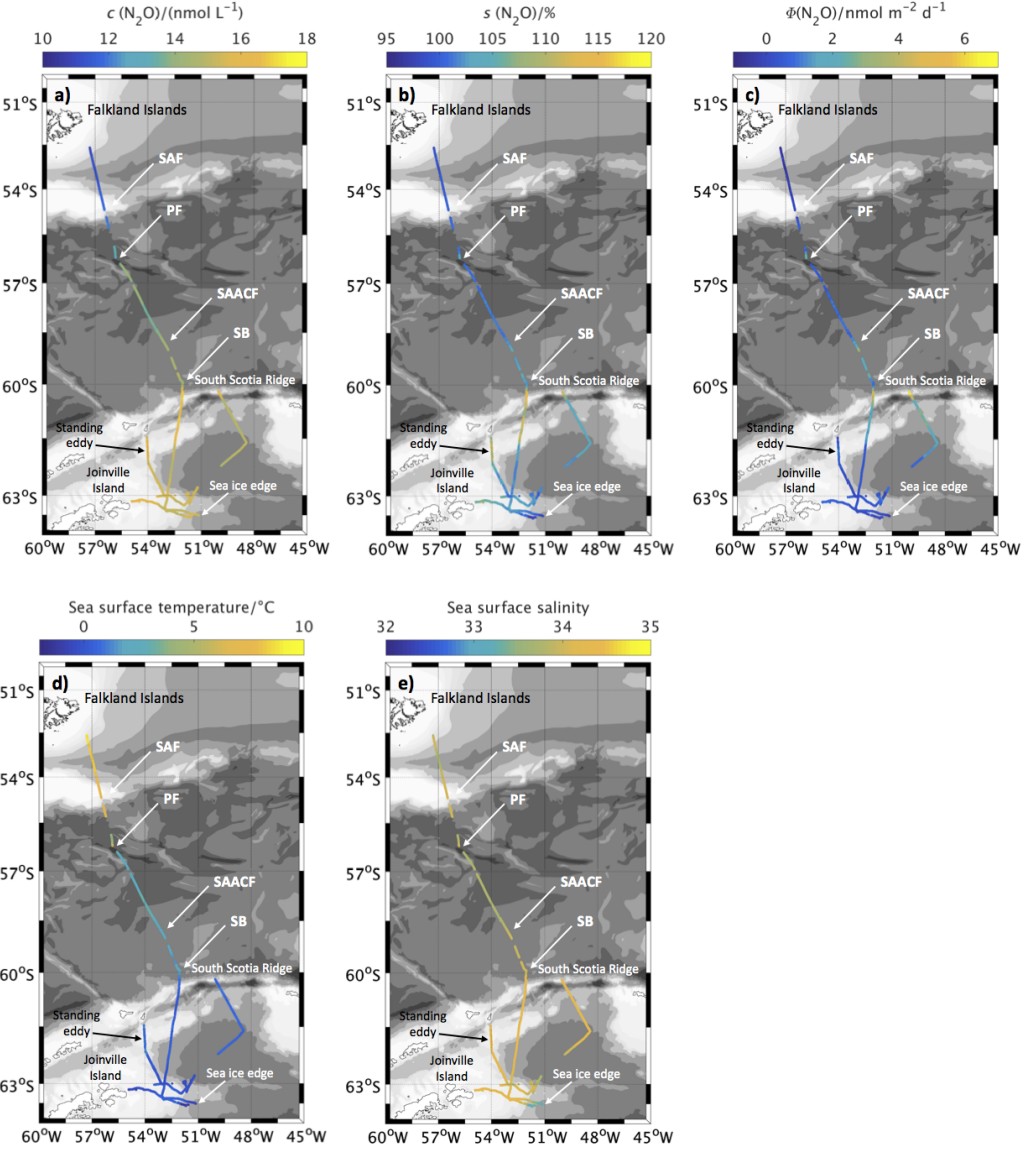

**Figure 5  N₂O surface concentrations, saturations and fluxes, as well as sea surface temperature and salinity along cruise track for JR255A.** (A) N₂O concentration in surface waters along the cruise track of JR255A. (B) N₂O saturations calculated from measured atmospheric and seawater dry mole fractions. (C) Sea-to-air flux along the cruise track of JR255A. (D) Sea surface temperature and (E) salinity from under-way measurements. Approximate position of Subantarctic Front (SAF), Polar Front (PF) and Southern Boundary (SB) are indicated, based on sea surface temperature and salinity measurements during JR255A.

Just south of the Southern Boundary of the Antarctic Circumpolar Current (SB), concentrations as well as saturations reached the highest values observed during JR255A/GENTOO. Temperature and salinity characteristics of the surface water indicated an influence of Upper Circumpolar Deep Water (UCDW) from below (Fig. 3B, *Orsi, Whitworth & Nowlin, 1995*). UCDW is an old water mass, high in remineralised nutrients and N₂O as a byproduct of nitrification. Depth profiles in the Indian sector of the Southern

Ocean displayed $N_2O$ maxima in the oxygen minimum layer of the UCDW and potential transport of these waters to the surface (*Chen et al., 2014*; *Zhan et al., 2015*). Entrainment of UCDW across the thermocline, as inferred from a decrease in surface temperature and an increase in salinity, could explain the high saturations across the SB during JR255A/GENTOO. Additionally, iron input from the Scotia Ridge, as observed by *Klunder et al. (2014)* could enhance *in situ* productivity, supplying substrate in form of sinking particles for *in situ* $N_2O$ production.

*Weddell sea.* Surface $N_2O$ concentrations of the open waters of the Weddell Sea were on average $(15.2 \pm 0.2)$ nmol $L^{-1}$ $((102.2 \pm 1.6)\%$ saturation). South of the ACC, the surface waters were colder than across Drake Passage, increasing the solubility of $N_2O$ and resulting in slightly lower saturations despite higher concentrations. Small but persistent supersaturations during JR255A were in contrast to $N_2O$ data from the Indian sector, where surface waters were mainly undersaturated (*Chen et al., 2014*; *Zhan & Chen, 2009*; *Zhan et al., 2015*). Meltwater low in $N_2O$ in combination with strong stratification, preventing exchange across the thermocline were identified as possible reasons for $N_2O$ undersaturations. While underaturations were observed during JR255A/GENTOO close to the sea ice edge (see below), meltwater does not seem to have a strong impact on the $N_2O$ saturations of surface waters in the Weddell Sea. Further studies are required to identify the differences between the Atlantic and Indian sector of the Southern Ocean in terms of meltwater contribution to the mixed layer and strength of stratification and mixing across the thermocline.

Higher-than-average $N_2O$ saturations for the Weddell Sea during JR255A/GENTOO were observed on the Antarctic shelf off the tip of Joinville Island $((104.2 \pm 1.7)\%)$ and during the section across the large standing eddy forming over the South Scotia Ridge, centred on 62°S and 54°W (*Thompson et al., 2009*) $((110.5 \pm 1.2)\%)$ (Figs. 5A and 5B). Iron from the sediments, as well as land run-off could have stimulated productivity in these areas (*Klunder et al., 2014*; *Sañudo Wilhelmy et al., 2002*; *Thompson & Youngs, 2013*; *Wadley, Jickells & Heywood, 2014*). In addition, the sampling region was more sheltered from the circumpolar winds by the Antarctic Peninsula, stabilising the water column over the shallow bathymetry. The combination of ample supply of trace nutrients and a stable water column, keeping phototrophic organisms in the euphotic zone, presumably resulted in comparably high productivity and subsequent $N_2O$ production during remineralisation of sinking biomass. In the vicinity of dynamic features, such as frontal zones, average chlorophyll concentrations for January 2012 show an increase in chlorophyll a over the shelf and across the standing eddy where $N_2O$ saturations are highest (Fig. 4: 25 and 29 Jan). In areas where physical transport and mixing is reduced, $N_2O$, as well as chlorophyll, can accumulate and show the expected relationship between high productivity and $N_2O$ production during nitrification as sinking biomass is remineralised.

*Sea ice edge.* Lowest saturations of on average $(98.8.0 \pm 1.3)\%$ were observed close to the sea ice edge in the south east of the survey region (Fig. 5B). Average $N_2O$ concentrations at the ice edge were $(15.2 \pm 0.2)$ nmol $L^{-1}$, similar to the average of $(15.4 \pm 0.2)$ nmol $L^{-1}$

observed in the other open ocean regions in the Weddell Sea. Saturations, however, were lower due to lower temperature and salinity, resulting in higher $N_2O$ solubility. *Randall et al. (2012)* observed under-saturations of $N_2O$ within sea ice due to loss of dissolved gases during brine rejection. Mixing of seawater with under-saturated melt water from sea ice that has undergone brine rejection during formation would decrease surface concentrations, which was not observed during JR255A. However, it is possible that end-members were not captured in these measurements and the mixing line for temperature and salinity indeed indicates dilution of Weddell Sea surface waters with melt water low in $N_2O$.

## Comparison of $N_2O$ surface saturations to chlorophyll a and nitrate concentrations

Satellite products and climatologies are readily available datasets, covering pigment and nutrient distribution for wide areas of the global ocean. To investigate potential links of $N_2O$ surface saturations to productivity and remineralisation, we have tried to find correlations between $N_2O$, chlorophyll *a* concentrations (as proxy for biomass accumulation) and nitrate concentrations (as proxy for integrated remineralisation). A tentative link between chlorophyll *a* concentrations from satellite products and $NO_3^-$ concentrations from climatologies could also open up these data sources to infer global and long-term $N_2O$ saturations without relying on *in situ* data or $N_2O$ models.

Surface saturations of $N_2O$ were compared to monthly average chlorophyll *a* concentration for December 2011 and January 2012 (Figs. 5 and 6). There was no simple correlation between $N_2O$ saturations and chlorophyll *a* concentrations, neither for specific areas of the two cruises, nor for the complete dataset (Figs. 6A and 6B). However, average chlorophyll was high in proximity to the crossings of the PFZ during JR260B (Fig. 4A: 30 Dec, 15 Jan).

In addition to satellite chlorophyll values, $NO_3^-$ concentrations from the World Ocean Atlas climatology (https://www.nodc.noaa.gov/OC5/woa13/woa13data.html) were calculated for the mixed layer (MLD, 60 m) and were compared to $N_2O$ saturations for JR260B and JR255A/GENTOO to test whether $NO_3^-$ as end-product of nitrification could be used as a proxy for mixed layer $N_2O$ production (Figs. 7A and 7B). However, again no clear correlation was observed between $N_2O$ saturations and $NO_3^-$ concentrations, neither for specific survey areas and oceanographic features, nor for the entire dataset.

Chlorophyll as a proxy for productivity and $NO_3^-$ as the endproduct of nitrification (a major $N_2O$-producing pathway) might be expected to show similar behaviour to $N_2O$ mixed layer saturations. However, there are reasons why we do not observe simple correlations between these parameters.

Firstly, there is the methodological issue of comparing near real-time $N_2O$ data with monthly averages (chlorophyll) and climatology data ($NO_3^-$). The location of frontal systems and upwelling water masses in the Southern Ocean are not fixed. Mixing and transport of water will influence observations during JR260B and satellite and climatology data.

Secondly, biomass (represented by chlorophyll) is not immediately available to nitrifying organisms. Breaking down of sinking biomass and release of $NH_4^+$, the substrate for nitrification, introduces a time delay for $N_2O$ production. $NH_4^+$ and $NO_3^-$ are not

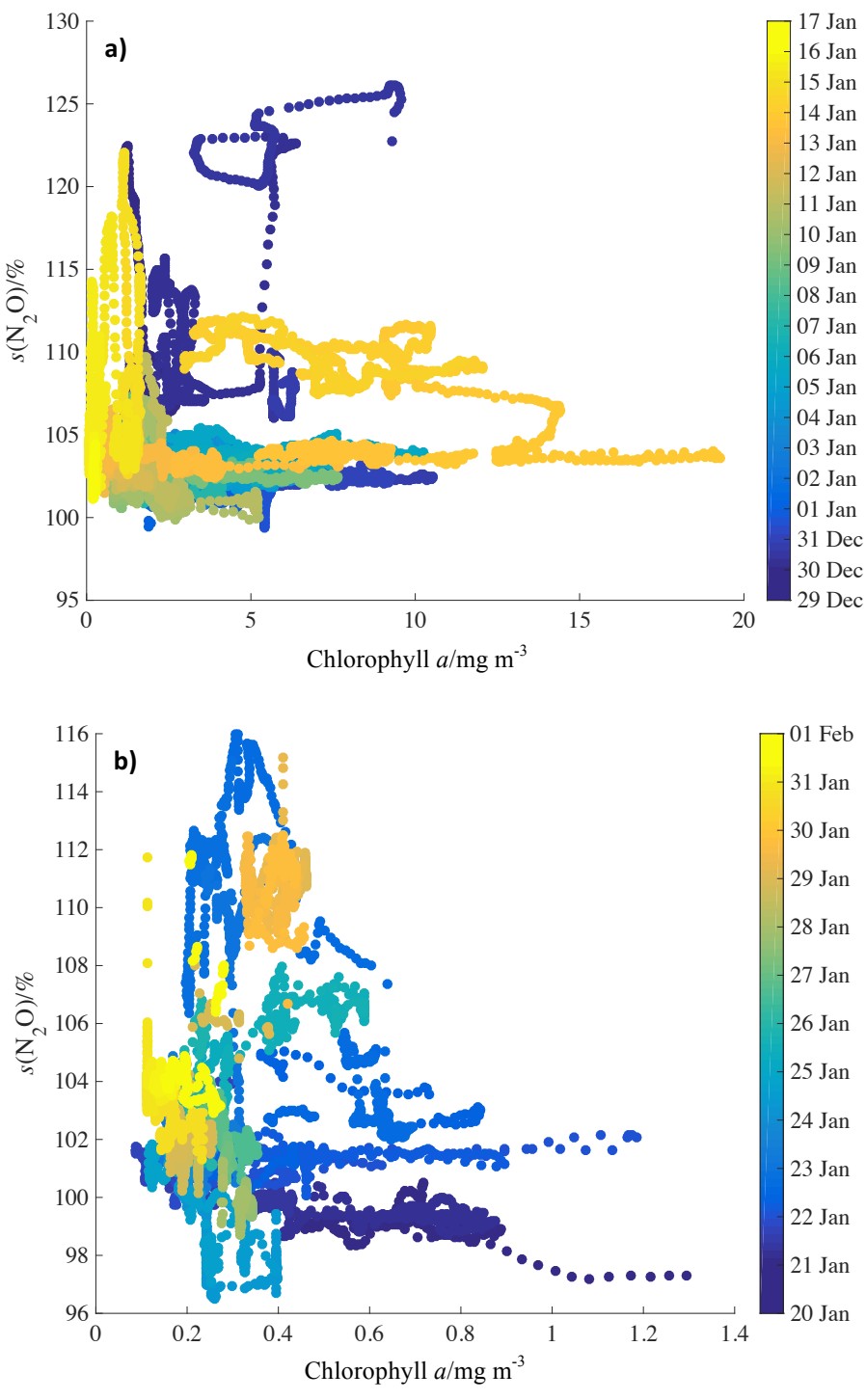

**Figure 6** **Relationship of N₂O saturations to chlorophyll concentrations by date.** N₂O saturations during (A) JR260B (December 2011) and (B) JR255A/GENTOO (January 2012) plotted against average satellite chlorophyll concentrations for December 2011 from MODIS Aqua. Symbols are colour-coded for date. Note the different axis scales.

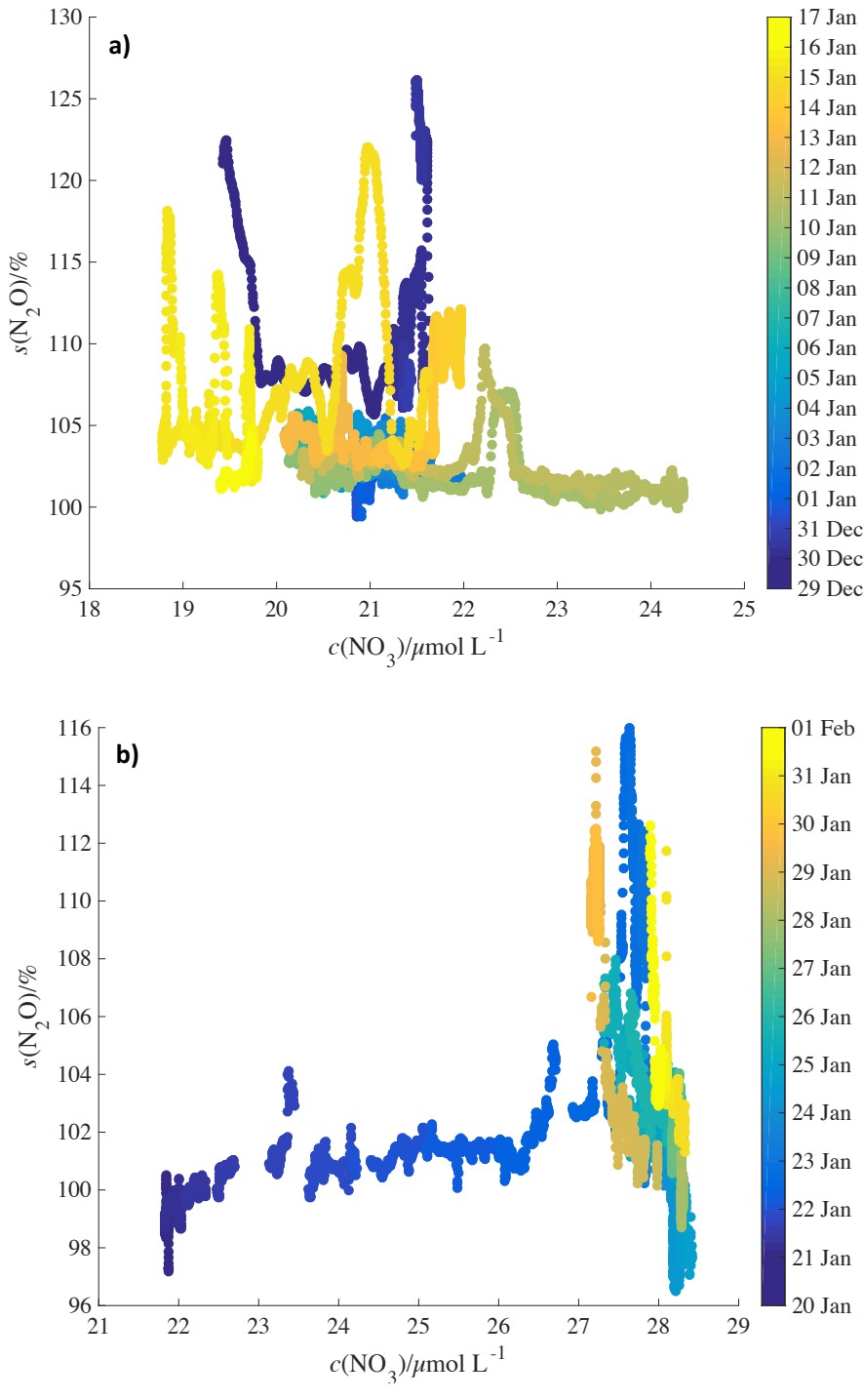

**Figure 7** **Relationship between N₂O saturations and MLD NO₃⁻ concentrations by date.** N₂O saturations (A) during JR260B and (B) JR255A/GENTOO plotted against mean winter MLD NO₃⁻ concentrations from World Ocean Atlas. Symbols are colour-coded for date.

exclusively controlled by the activity of nitrifying bacteria and archaea. They are also important nutrients for phytoplankton, therefore uptake by algae is further complicating the use of $NO_3^-$ as a proxy for $N_2O$ saturations.

The lack of a clear, quantitative link between $N_2O$ saturations and observations of chlorophyll *a* and $NO_3^-$ makes it difficult to infer $s(N_2O)$ (unlike $CO_2$) by multi-linear regression (*Wallace, 1995*) or prognostic neural network models (*Landschützer et al., 2014*). Despite the absence of a quantitative correlation, high $N_2O$ saturations were accompanied by an increase in chlorophyll *a* concentrations on the Antarctic shelf and across the standing eddy.

## $N_2O$ sea-to-air flux

Sea-to-air flux during JR260B and JR255A/GENTOO was calculated as described above. For comparison to global average sea-to-air flux, a global ocean area of $361.9 \times 10^6$ km$^2$ was used (*Eakins & Sharman, 2010*). Oceans contribute approximately 25% to global $N_2O$ emissions of 17.9 Tg a$^{-1}$ (N equivalents) (*Ciais et al., 2013*), resulting in a marine source of 4.5 Tg a$^{-1}$ or $4.4 \times 10^8$ mol d$^{-1}$. This results in an average sea-to-air flux of 1.2 $\mu$mol m$^{-2}$ d$^{-1}$ for the global ocean. For comparison, *Nevison, Weiss & Erickson (1995)* estimated a global oceanic $N_2O$ source of 4 Tg a$^{-1}$, resulting in a slightly lower average flux of 1.08 $\mu$mol m$^{-2}$ d$^{-1}$.

### JR260B

Surface waters during JR260B were mainly a source of $N_2O$ to the atmosphere (Fig. 2C). The average flux of 2.2 $\mu$mol m$^{-2}$ d$^{-1}$ was almost twice the global average flux of 1.2 $\mu$mol m$^{-2}$ d$^{-1}$, based on *Ciais et al. (2013)*.

The average sea-to-air flux within the SAZ was $(1.1 \pm 0.4)$ $\mu$mol m$^{-2}$ d$^{-1}$, due to low to moderate wind speeds at the time of measurements and saturations just slightly above equilibrium with the atmosphere. The data from the SAZ were limited to the very beginning and very end of each transit to the Scotia Sea and Weddell Sea, therefore the flux values presented here might not be representative of the wider SAZ. Despite the proximity to the Falkland Islands, no enhancing island effect on $N_2O$ saturations and flux was observed (in comparison to South Georgia). While this work is focussed on emissions across the ACC and south of the SB, more data are required from subpolar regions for a better understanding of oceanic $N_2O$ sources.

$N_2O$ flux progressively increased across the PFZ to an average of $(3.6 \pm 2.4)$ $\mu$mol m$^{-2}$ d$^{-1}$ with highest values of 13.2 $\mu$mol m$^{-2}$ d$^{-1}$ during transit from South Georgia back to the Falkland Islands. Average global flux values were highly exceeded across the frontal zone as high $N_2O$ supersaturations, possibly a result of *in situ* production due to high biomass supported by iron supply from sediments, coinciding with high circumpolar wind speeds. As these high emissions were not observed during the crossing of the PFZ further to the west during JR255A/GENTOO (see below), it would be important to find out if this is a local phenomenon, influenced for example by the deflection of the ACC around South Georgia, or if the PFZ is a strong source of $N_2O$ to the atmosphere in other sectors of the Southern Ocean. Bathymetry might be a controlling factor as a source of iron to the water column, supporting productivity and subsequent $N_2O$ production during remineralisation (*Wadley, Jickells & Heywood, 2014*).

Sea-to-air flux decreased again in the open waters of the AAZ around South Georgia as $N_2O$ supersaturations were lower. The average flux of $(1.8 \pm 1.8)$ µmol m$^{-2}$ d$^{-1}$ was, however, still exceeding the global average. As productivity in the Scotia Sea is higher than in other HNLC areas of the Southern Ocean, these high fluxes might be restricted to similar areas where islands presumably enhance productivity and $N_2O$ emissions. Negative flux up to -0.5 µmol m$^{-2}$d$^{-1}$ was only observed in a small area to the east of the PF during transit to South Georgia. It was not apparent what caused this localised and potentially transient sink; uptake by nitrogen fixing bacteria might be a possible mechanism for loss of $N_2O$ from surface waters (*Farías et al., 2015*).

In contrast to the open waters of the AAZ, Stromness Bay was a strong source of $N_2O$ to the atmosphere while the ship was anchored for calibration of acoustic instruments. Sea-to-air flux was initially only slightly higher than for the surrounding open ocean area (2.2 µmol m$^{-2}$ d$^{-1}$), despite high $N_2O$ supersaturations in the bay (on average around 108.1%). This was due to low wind speeds of 7.3 m s$^{-1}$, which subsequently increased to 14.5 m s$^{-1}$ over 25 h, resulting in sea-to-air fluxes of up to 10.6 µmol m$^{-2}$ d$^{-1}$ and an average of $(4.5 \pm 2.4)$ µmol m$^{-2}$ d$^{-1}$. These observations highlight the impact of island productivity, as well as changes in the weather/wind speed for $N_2O$ emissions from the ocean. While the coastal region of South Georgia only accounts for a small surface area of the Scotia sea, emissions are very high. If this effect could be observed across other island systems in the Southern Ocean, the cumulative effect might be of significance for global flux estimates. Sea-to-air flux was controlled by increasing wind speed (saturations were stable for the 25 h observation period), which highlights the impact of changes to the climate system for the source strength of regions with high surface $N_2O$ saturations. This is of particular importance for the Southern Ocean where an increase in wind speed is expected for the future (*Young, Zieger & Babanin, 2011*; *Hande, Siems & Manton, 2012*).

Overall, the surface ocean across the Scotia Sea was a strong source of $N_2O$ to the atmosphere for most of the research cruise JR260B with sea-to-air fluxes exceeding global average flux. These high fluxes were driven by high surface water saturations, presumably resulting from *in situ* production, and moderate to high wind speed.

### JR255A/GENTOO

The average sea-to-air $N_2O$ flux throughout JR255A/GENTOO was $(0.6 \pm 0.9)$ µmol m$^{-2}$ d$^{-1}$, which is below global average values of 1.2 µmol m$^{-2}$ d$^{-1}$ and considerably lower than the average flux for JR260B, just to the north of the JR255A/GENTOO region. The low flux was a result of lower saturation values compared with JR260B, but still above equilibrium with the atmosphere for most of the cruise. Additionally, wind speeds were lower during JR255A/GENTOO.

The highest sea-to-air flux of 6.7 µmol m$^{-2}$ d$^{-1}$ was observed across the South Scotia Ridge where high wind speeds coincided with high $N_2O$ supersaturations, presumably due to surface waters being influenced by underlying UCDW high in $N_2O$. For a better estimate of Southern Ocean $N_2O$ emissions, it would be important to investigate the circumpolar source strength of $N_2O$ along the SB where UCDW is expected to influence the SML. In combination with potentially increasing wind speed in the Southern Ocean (*Young, Zieger*

& *Babanin, 2011*; *Hande, Siems & Manton, 2012*), the SB could be a substantial source of $N_2O$ to the atmosphere.

Although surface waters were most strongly undersaturated at the sea ice edge, negative fluxes were highest at the beginning of the cruise, close to the Falkland Islands ($-1.3$ $\mu$mol $m^{-2}$ $d^{-1}$). Highest wind speeds of up to 13.3 m $s^{-1}$ were observed at that time, driving the strong negative fluxes on the shelf. At the ice edge, in contrast, wind speeds were considerably lower (1.4 to 6 m $s^{-1}$), resulting in weaker negative fluxes. The underlying mechanism for undersaturation on the Falkland shelf is currently unknown, but might be related to seasonal heat fluxes/cooling of the surface ocean, which occurs on faster time scales than air-sea gas exchange. Similar to localised negative fluxes during JR260B, uptake of $N_2O$ as an alternative substrate during biological nitrogen fixation could also be a sink (*Farías et al., 2015*).

$N_2O$ sinks at the sea ice edge are most likely controlled by physics by increasing solubility in cold, fresh waters or by $N_2O$ drawdown through brine rejection (*Randall et al., 2012*). Physical drawdown of $N_2O$ could buffer the atmospheric increase, but water high in $N_2O$ would nevertheless eventually find its way back into the surface mixed layer. The impact of the sea ice edge sink on oceanic $N_2O$ emissions is currently not very well understood and requires further observations, as well as integration into emission models.

Low wind speeds also affected sea-to-air flux on the Antarctic shelf close to Joinville Island, as well as across the standing eddy over the South Scotia Ridge. Despite relatively high supersaturations, $N_2O$ flux was rather low (($0.4 \pm 0.2$) and ($0.5 \pm 0.4$) $\mu$mol $m^{-2}$ $d^{-1}$, respectively). As the data collected during JR255A/GENTOO only present a snapshot of $N_2O$ saturations and resulting sea-to-air flux, shelf areas and eddies might contribute more strongly to global emissions at other times. Productivity is expected to be high throughout the summer months, and a change in weather and wind speed could substantially increase the source strength of these areas (see JR260B, Stromness Bay above). Again, additional data throughout the year, as well as from other sectors of the Southern Ocean, are required to better understand the strength and variability of $N_2O$ emissions. Regional biogeochemical and emission models might be helpful to elucidate the links between productivity, dominant nitrogen and $N_2O$ cycling pathways and changing weather conditions to improve estimates for the Southern Ocean source strength.

Throughout JR255A/GENTOO sea-to-air flux of $N_2O$ was low compared to JR260B, as well as the global average, mainly due to low wind speed rather than low $N_2O$ saturation of surface waters. The continental shelf and oceanographic features like eddies hold the potential for substantial $N_2O$ sources if wind speed increases and production in the water column is sustained.

## CONCLUSIONS

High-resolution (1 min averages of measurements at 1 Hz) $N_2O$ data were collected from the surface waters of the Atlantic sector of the Southern Ocean during transit and station work, covering highly varied areas across the ACC and the AAZ. These data from the austral summer 2011/2012 add substantially to the limited observations of $N_2O$ emissions from

the Southern Ocean, increasing observations from the Scotia Sea by 20,674 values (188 data points from 1989 previously published by *Weiss et al. (1992)*) and by 9730 from the Weddell Sea (250 data points from 1984 previously published by *Weiss et al. (1992)*), and show unprecedented fine-scale variability compared to the existing data (supplementary Figures 1 and 2 in *Weiss et al., 1992*).

Both, the Scotia Sea and the Weddell Sea, were a source of $N_2O$ to the atmosphere. The source strength was highly variable in space and time with rare areas of negative sea-to-air flux.

The following areas were identified as strong $N_2O$ sources by high sea-to-air flux at the time of the survey and/or high supersaturations:

- shallow bathymetry/ridges: North and South Scotia Ridge,
- shelf and coastal areas: The Antarctic shelf and coastal South Georgia and
- the Polar Frontal Zone: To the east of Drake Passage,

whereas $N_2O$ sources were associated with water mass transport at

- the Southern Boundary of the ACC, where shoaling UCDW was influencing surface waters of the mixed layer.

The following areas were identified as sinks for atmospheric $N_2O$ by negative sea-to-air flux at the time of the survey:

$N_2O$ sinks through physical processes:

- Sea ice edge: $N_2O$ sink due to increased solubility or brine rejection.

$N_2O$ sinks through other processes, presumably seasonal cooling and temporal lag between heat and gas fluxes and/or biological nitrogen fixation:

- Falkland shelf to the south of the islands.
- To the east of the PF in the Scotia Sea.

Neither chlorophyll *a*, nor $NO_3^-$ concentrations showed any correlation to $N_2O$ saturations. Assuming that the average flux values of 2.3 $\mu$mol m$^{-2}$ d$^{-1}$ and 0.7 $\mu$mol m$^{-2}$ d$^{-1}$ calculated for JR260B and JR255A are representative for the Scotia Sea and Weddell Sea over time, the combined area would contribute 0.04 Tg a$^{-1}$ N (nitrogen equivalent) to the global $N_2O$ source. While this value is relatively low, also compared to model estimates of 0.9 Tg a$^{-1}$ N (nitrogen equivalents) for the entire Southern Ocean (*Nevison et al., 2005*), areas with a strong variability in $N_2O$ saturation and sea-to-air flux were observed for both cruises. Short-term average flux for the Scotia Sea was almost twice as high as the global average. These findings indicate the importance of high-resolution data to accurately estimate the source strength of mesoscale features, such as frontal systems and eddies. This study highlights the need for further high-resolution data from other sectors of the Southern Ocean to resolve variability in $N_2O$ emissions and improve global emission estimates.

## ACKNOWLEDGEMENTS

We would like to thank two anonymous reviewers and the editor for their helpful comments which greatly improved the manuscript. Thanks to the officers and crew on board RRS James Clark Ross and the scientific party for their support during JR260B and JR255A/GENTOO. We also thank Sunke Schmidtko for supplying calibrated sea surface temperature and salinity, Dorothee Bakker for providing the equilibrator used in this study and Gareth A. Lee for technical support. $N_2O$ data from *Weiss et al. (1992)* was downloaded from the MEMENTO database (https://memento.geomar.de/; *Kock & Bange, 2015*).

### Funding

This study was supported by the European Community's Seventh Framework Programme (FP7/2007-2013) under grant agreement number 237890 (Marie Curie Initial Training Network "INTRAMIF"), the NERC Collaborative Gearing Scheme, project number AFI CGS78 and the BAS Ecosystems Long Term Monitoring and Survey Programme Western Core Box (LTMS WCB). The GENTOO project and cruise were supported by NERC Antarctic Funding Initiative grant NE/H01439X/1. The funders had no role in study design, data collection and analysis, decision to publish, or preparation of the manuscript.

### Grant Disclosures

The following grant information was disclosed by the authors:
European Community's Seventh Framework Programme (FP7/2007-2013): 237890.
AFI: CGS78.
BAS Ecosystems Long Term Monitoring and Survey Programme Western Core Box (LTMS WCB).
NERC Antarctic Funding Initiative: NE/H01439X/1.

### Competing Interests

Jan Kaiser is an Academic Editor for PeerJ.

### Author Contributions

- Imke Grefe conceived and designed the experiments, performed the experiments, analyzed the data, contributed reagents/materials/analysis tools, prepared figures and/or tables, authored or reviewed drafts of the paper, approved the final draft.
- Sophie Fielding and Karen J. Heywood analyzed the data, contributed reagents/materials/analysis tools, authored or reviewed drafts of the paper, approved the final draft.
- Jan Kaiser conceived and designed the experiments, analyzed the data, contributed reagents/materials/analysis tools, authored or reviewed drafts of the paper, approved the final draft.

## Data Availability

Data has been deposited at the MEMENTO database (MarinE MethanE and NiTrous Oxide) and is available upon request: https://memento.geomar.de/database. Data can also be found in the Supplemental Information.

## Supplemental Information

Supplemental information for this article can be found online at http://dx.doi.org/10.7717/peerj.5100#supplemental-information.

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
