# Peer review of "Nitrous oxide variability at sub-kilometre resolution in the Atlantic sector of the Southern Ocean"

_PeerJ, doi:10.7717/peerj.5100_

## Round 0.1 · original submission · Major Revisions

We have received the reports from our advisors on your manuscript, "Nitrous oxide variability at sub-kilometre resolution in the Atlantic sector of the Southern Ocean", submitted to PeerJ

Based on the advice received, I have decided that your manuscript requires corrections as suggested by the reviewer(s) before it can be accepted for publication.

My main concern deals with the drivers for the N2O supersaturation measured during the cruises, which is solely attributed to nitrification

Reviewer 1 ·

Basic reporting

no comment

Experimental design

no comment

Validity of the findings

no comment

Additional comments

General commend
N2O is one of the most important greenhouse gases; it also acts as key ozone depletor. The Southern ocean was considered as one of the most important N2O source to the atmosphere; however, very limit data is available. The authors provide a set of precious data using a state-of-the-art technique. The data is in good quality, and provide some very important information, base on which the authors draw reasonable conclusions. I understand the difficulty of writing a manuscript using a single set of N2O data. However, it seems that the authors do not make good use of their surface hydrographic data. The authors separate the study area in the study area, but the criteria for this separation, and the hydrographic processes that influence the distribution of these distribution patterns is not clearly addressed; instead, the authors discussed the factors that regulate the distribution patterns by citation of other studies, but some of the citation is inappropriate, and the mechanism of how the source and sink characteristic are ambiguous. Moreover, the structure of the MS may need to be rearranged. Major revision is needed before this paper can be accepted for publication.





Specific commend

Line 147. The air sea exchange evaluation method of Wanninkhof 1992 is widely used, but Wanninkhof 2014 published another work to update the 1992 result, which I would suggest the authors to use.

Line 154. JR260B and JR255A/GENTOO are two cruise tracks both cross different frontal structures of the southern ocean, there may be possible similar phenomenon along this cruise tracks, whereas the JR260B itself contains a round trip cruise track, also show some different signal on the same region, that can also provide some interesting information.

Line 166. How the authors determine the location of the frontal structure? Generally, there are different criteria that mark the location of the frontal structures, including temperature, salinity or oxygen at certain depth. The authors should provide these criteria for different fronts.

Line 200. There is not enough evidence to convince the reader about possible remineralization of sinking bloom biomass. There is a gap between the de Baar’s work and the above conclusion. A image or description of distribution of chlorophyll distribution pattern may be helpful.

Line 223. Antarctic Surface Water should be deleted

Line 237. One of my concerns is why the authors arrange the sections in this way, they discuss the distribution of N2O and saturation, then flux, and then back to possible biogeochemical processes that may influence the distribution? It will be better to discuss all the possible factors that will influence the distribution pattern, then to discuss the result of the air-sea flux evaluation.

Line 255. The citation of Bange et al (1996) used here may be inappropriate, these are two very different locations.

Line 267. Again, the criteria of these frontal structures are needed.

Line 295. Is the nutrient or Dissolved oxygen measured during this cruise, especially the latter, which is an important criterion to indicate the UCDW?

Line 344. I don’t think undersatrautions of N2O in the sea ice appropriate expression, there is no study about the solubility of N2O in the sea ice. A quote here may be more rigorous.

Line 385. It’s inappropriate to consider this -0.5 μmolm-2 d-1 as a sink, even the authors does not provide the uncertainty of this flux. Therefore, the possible reason of the “sink” should be cautious.

Line 441 the term “Eddies” appears for the first time in the MS after the Abstract, it also appears in the Figures 3. Is there more information about this eddy, how to identify this eddy structure, and what kind of eddy? At least the authors should provide this information in results.

Line 456. As mentioned before, I wonder why the authors put this section as a final part of this MS. There are factors those will influence the distribution of the N2O and so as to it air-sea flux, including hydrographic ( temperature, salinity, frontal structures) and the biogeochemical process, which can be learn from distribution patterns of nutrient and chlorophyll etc. so, it will be better to discuss all the factors influence the distribution then to discuss the air sea flux.

Line 457. Did the authors try to plot the N2O against NO3- or Chlorophyll in different frontal zone or front respectively? It’s very difficult to find correlation between N2O and Chlorophyll or NO3- of the whole study area, since different hydrographic and biogeochemical processes presents in the different zone or front. For example, ASSW is fresh and cool water mass which should be more equilibrate with the atmosphere, whereas where upwelling of CDW approaching the surface, higher N2O concentration presents, then above correlation should definitely be different.

Line 476 superscript of NO3- is missing, and there are several same typo below.

Line 515 The term “sink” should be used carefully, the “sinks” generally present during summer when the sea ice melt or surface water cooling, therefore it should be seasonal event, so temporal sinks is a more appropriate expression than “sink”.


About the Figures
Figure 1. I would like to know how the lines marks the frontal structure comes, satellite remote sensing data?

Figure 2. I would suggest the authors to expand the color scales of all these Figures (the same for Figure 3 and Figure 4), that may provide more information for this Figure.

Figure 3. The T-S plots can provide important information about the water masses; however, for the surface water it will not be that effective. Moreover, I think it is inappropriate that the authors mark the “approximate” positions of the fronts in Figure 3. The frontal structure generally shows sharp change of parameters such as temperature, salinity etc. Take SAF for example, which marks by sudden change of temperature from 4-8°C. Therefore, its hard marks the approximate position of the SAF on the ST plot. Instead of using ST plot, I suggest the authors use latitude/longitude/distance against N2O concentration/Saturation/temperature/Salinity, with which, frontal structure will be easier to indicate.

Reviewer 2 ·

Basic reporting

The manuscript is generally well written and structured. The introduction shows necessary information on the N2O distribution in the Southern Ocean with appropriate literature cited. Some more information on the microbial (nitrification, denitrification, nitogen fixation) and physical processes (transport processes, solubility effects) that can influence the N2O distribution should be given, however, as these are discussed in the course of the manuscript.

The manuscript is accompanied by the measured concentrations of N2O, its saturation and corresponding fluxes, together with sampling time and position data in the supplement. The data seem sound and presented in a clear format; it would nevertheless be very helpful to also include temperature and salinity as well as wind speed in the data submission. This would allow the recalculation of saturation and flux measurements.

Experimental design

The manuscript „Nitrous oxide variability at sub-kilometre resolution in the Atlantic sector of the Southern Ocean“ by Grefe et al. reports surface measurements of nitrous oxide from two cruises in the Southern Ocean. These data help to fill an important gap in global ocean N2O flux estimates, as measurements from the Southern Ocean are sparse and its role for N2O emissions is still unclear.

The method description of the N2O measurements and the calculation of saturation and air-sea fluxes is detailed and comprehensive. However, I found it confusing that the nitrate and chlorophyll a data products that were used are described quite at the end of the manuscript (section 3.3), but potential links between N2O and chlorophyll a discussed already in earlier sections. Maybe it makes sense to move the description of the data products to the methods section. I would also recommend to include more details on the data products used – e.g. which spatial resolution was used? How were the N2O data matched onto the gridded data products (see my comment below)?

Specific comments:

Lines 143-144: please specify how the in-situ data are matched to the CCMP data. Did you use the closest (in time and space) available data or were the CCMP data interpolated? Were CCMP data compared to in-situ wind speeds?

Lines 159-160: „Surface water saturations [...] based on these values.“ Did you use the average atmospheric mole fractions from the respective cruises (323.6 and 324.0) to calculate N2O saturations and fluxes?

Validity of the findings

While the N2O distribution and the derived N2O emissions seem conclusive, the discussion of the N2O source processes needs some improvement:
The authors attribute the general N2O supersaturation found during their measurement campaings mainly to N2O production by nitrification and try to correlate N2O saturations with chlorophyll a and surface concentrations of nitrate as indicators for nitrification. I have several difficulties with this argumentation:

I) Other processes that could influence N2O saturations in the surface waters should be discussed in more detail. Can physical processes account for the observed saturations (e.g. cooling/warming of surface waters, mixing of water masses with different temperatures/salinities, bubble effects) What is the role of diapycnal mixing for the generation of oversaturation in surface waters? Rees et al. (1997) reported substantial oversaturations of N2O in the circumpolar deep water, a finding which is consistent with the increase in N2O in waters influenced by the UCDW found during JR255A/GENTOO. Could these waters supply sufficient N2O to the mixed layer to generate the observed saturations?

II) I doubt that the correlation between the in-situ N2O data and monthly averaged Chla and climatological nitrate for the entire dataset provides much meaningful information (as shown in Figures 5, 6 and 7. Surface chlorophyll a as well as nitrate concentrations can at best be interpreted as indirect indicators for nitrification and associated N2O production, as nitrification is attributed to remineralization, not to primary production. This process may not coincide in time and space with high productivity.
Similarly, high nitrate concentrations in the surface ocean are not necessarily a good indicator for nitrification, as nutrient assimilation strongly influences the surface nitrate concentrations.

Moreover, as mentioned by the authors themselves, there may be a significant bias between in-situ measurements and monthly averaged or even climatological datasets. Nevertheless there may be a correlation between N2O and chlorophyll a for parts of the cruise track (e.g. the data from Stromness Bay) as an indication for an influence of primary productivity on N2O emissions.

Additional comments

no comment

---

## Round 0.2 · Minor Revisions

I find that the points raised by the two reviewers have been adequately dealt with. However, there are still some minor points that require revisions. In particular, the correlation between N2O data and biogeochemical parameters (e.g. chlorophyll and nitrate concentration)
could be better explored and commented.

Reviewer 1 ·

Basic reporting

no comment

Experimental design

no comment

Validity of the findings

no comment

Additional comments

This paper is improved alot after rivision, though its a pity that no enough matching parameters like nutrient and DO and so on for each cruise, which can not be improved by further revision, however, the N2O underway data in this region is an important contribution to community. Therefore, I recommend publish of this paper, but before that some minor revision is need.
I would like to see the author provide the criteria of the frontal structure in the paper. The authors response that there are criteria explained in section 3.1.1, they surely are, however, the explanation is something like warmer and colder, or freshening, that's quanlitative discription not quantitative, if possible a temperature or salinity range as a criteria will be better, the reader may not familiar with the proerties of these frontal structure, the may not be able to tell them from the figures.

Reviewer 2 ·

Basic reporting

Please view my previous review and my comments to the authors below.

Experimental design

Please view my previous review and my comments to the authors below.

Validity of the findings

Please view my previous review and my comments to the authors below.

Additional comments

The revised manuscript was improved by the authors. As suggested, information on the physical and biological processes influencing N2O concentrations in seawater have been added to the introduction section, and the data accompanying the manuscript now allow the re-calculation of saturation and gas exchange fluxes. Likewise, information on the methods used for matching the gridded Chlorophyll a and nitrate data products was added.

I think that the data presented in this manuscript are a great contribution to the knowledge on the oceanic N2O distribution.

My main concern regarding this manuscript however was that the authors try to find a correlation between N2O saturation and Chlorophyll a and nitrate concentrations as an indicator for in-situ N2O production. In the manuscript, the authors indeed discuss the shortcomings of this approach. I would however be interested if at least short-term data products of satellite Chlorophyll a (e.g. 3-day or 8-day averages) are available that could be matched to the N2O data and produce more significant correlations between these parameters.

---

## Round 0.3 · accepted · Accept

We appreciate your patience and we hope that this improved publication structure will encourage you to submit your latest research results again PeerJ